# *In Silico*, *In Vitro* and *In Cellulo* Models for Monitoring SARS-CoV-2 Spike/Human ACE2 Complex, Viral Entry and Cell Fusion

**DOI:** 10.3390/v13030365

**Published:** 2021-02-25

**Authors:** Delphine Lapaillerie, Cathy Charlier, Henrique S. Fernandes, Sergio F. Sousa, Paul Lesbats, Pierre Weigel, Alexandre Favereaux, Véronique Guyonnet-Duperat, Vincent Parissi

**Affiliations:** 1Fundamental Microbiology and Pathogenicity Lab (MFP), UMR 5234 CNRS-University of Bordeaux, SFR TransBioMed, 33076 Bordeaux, France; delphine.lapaillerie@u-bordeaux.fr (D.L.); paul.lesbats@u-bordeaux.fr (P.L.); 2IMPACT Platform “Interactions Moléculaires Puces ACTivités”, UMR CNRS 6286 UFIP, Université de Nantes, F-44000 Nantes, France; cathy.charlier@univ-nantes.fr (C.C.); pierre.weigel@univ-nantes.fr (P.W.); 3UCIBIO@REQUIMTE, BioSIM -Departamento de Biomedicina, Faculdade de Medicina da Universidade do Porto, Alameda Professor Hernâni Monteiro, 4200-319 Porto, Portugal; hfernandes@med.up.pt (H.S.F.); sergiofsousa@med.up.pt (S.F.S.); 4Interdisciplinary Institute for Neuroscience (IINS-CNRS UMR 5297), Centre Broca Nouvelle Aquitaine, 33076 Bordeaux, France; alexandre.favereaux@u-bordeaux.fr; 5Vect’UB, vectorology platform, INSERM US05—CNRS UMS 3427-TBM-Core, Université de Bordeaux, 33000 Bordeaux, France; veronique.guyonnet-duperat@inserm.fr

**Keywords:** SARS-CoV-2, COVID, Spike/ACE2 complex

## Abstract

Severe acute respiratory syndrome coronavirus 2 (SARS-CoV-2) is the etiologic agent responsible for the recent coronavirus disease 2019 (COVID-19) pandemic. Productive SARS-CoV-2 infection relies on viral entry into cells expressing angiotensin-converting enzyme 2 (ACE2). Indeed, viral entry into cells is mostly mediated by the early interaction between the viral spike protein S and its ACE2 receptor. The S/ACE2 complex is, thus, the first contact point between the incoming virus and its cellular target; consequently, it has been considered an attractive therapeutic target. To further characterize this interaction and the cellular processes engaged in the entry step of the virus, we set up various *in silico*, *in vitro* and *in cellulo* approaches that allowed us to specifically monitor the S/ACE2 association. We report here a computational model of the SARS-CoV-2 S/ACE2 complex, as well as its biochemical and biophysical monitoring using pulldown, AlphaLISA and biolayer interferometry (BLI) binding assays. This led us to determine the kinetic parameters of the S/ACE2 association and dissociation steps. In parallel to these in vitro approaches, we developed in cellulo transduction assays using SARS-CoV-2 pseudotyped lentiviral vectors and HEK293T-ACE2 cell lines generated in-house. This allowed us to recapitulate the early replication stage of the infection mediated by the S/ACE2 interaction and to detect cell fusion induced by the interaction. Finally, a cell imaging system was set up to directly monitor the S/ACE2 interaction in a cellular context and a flow cytometry assay was developed to quantify this association at the cell surface. Together, these different approaches are available for both basic and clinical research, aiming to characterize the entry step of the original SARS-CoV-2 strain and its variants as well as to investigate the possible chemical modulation of this interaction. All these models will help in identifying new antiviral agents and new chemical tools for dissecting the virus entry step.

## 1. Introduction

Severe acute respiratory syndrome coronavirus 2 (SARS-CoV-2) is a positive-sense RNA virus that causes a severe respiratory syndrome in humans. It is responsible for the recent outbreak of coronavirus disease 2019 (COVID-19). To date, no treatment is available; therefore, a better understanding of the replication processes and the search for new antiviral approaches remain crucial. As the first contact point between the virus and the cell, the viral entry step constitutes an attractive target for blocking the very earliest events of viral infection.

Virus entry into cells requires the interaction between the spike S protein of the virus and its cellular receptor, angiotensin-converting enzyme 2 (ACE2). After binding to its cellular receptor, S protein proteolysis by host proteases (as trypsin and furin) occurs leading to the formation of S1 and S2 subunits. The further cleavage of the S2 domain releases the fusion peptide that triggers the activation of the membrane fusion allowing the ingestion of the virus by the cell and the next replication steps [1,2]. 

ACE2 [3] has been reported as a receptor for SARS-CoV-2 in the lung, as previously shown for SARS-CoV, which was responsible for the first SARS outbreak in 2003 [4]. ACE2 is expressed in human airway epithelia, lung parenchyma, vascular endothelia, kidney cells and small intestine cells but is only weakly detected in the brain [5,6]. However, in addition to respiratory syndrome, neurologic signs have been frequently observed, suggesting that the virus may also infect or invade the central nervous system (CNS) as supported by previous reports of CNS infection by SARS-CoV and the presence of viral particles in patient brains [7,8]. Taken together, these data strongly suggest additional routes for viral entry to the CNS. The results from the literature confirmed the neuronal invasion property of the virus as well as the requirement for the ACE2 receptor in this process [9,10]. Taken together, these data indicate that the S/ACE2 interaction remains the major entry pathway for SARS-CoV-2.

To further investigate this interaction and thus identify new antiviral agents as well as new chemical tools for dissecting this entry step, we set up *in silico, in vitro* and *in cellulo* systems to monitor the S/ACE2 association. We report here computational visualization and analysis of the S/ACE2 complex as well as its monitoring using biochemical pulldown methods and biophysics approaches as well as AlphaLISA and biolayer interferometry (BLI) binding assays. In addition to these *in silico* and *in vitro* approaches, we also developed a SARS-CoV-2 lentivirus infectivity assay and performed cell imaging and flow cytometry quantification of the S/ACE2 interaction. In addition to allowing the determination of the kinetic parameters of the interaction, our work provides a combination of methods for further analyses of the viral entry step and for the selection of drugs targeting this stage of the infection

## 2. Materials and Methods

### 2.1. In Silico Molecular Dynamics Simulation of the SARS-CoV-2 S/ACE2 Complex

A model was prepared considering the receptor binding domain (RBD) of the wild type Spike protein S1 of SARS-CoV-2 complexed with human ACE2. The X-ray structure with the PDB ID 6M0J, with a resolution of 2.45 Å was used [11]. This structure contains the RBD of S1 and includes all the amino acid residues of the RDB, with well-defined packing between the residues on S1 RDB and ACE2. In addition, the following N-glycosylation sites were identified and were taken into account: Asn53, Asn90, Asn322 and Asn546 in ACE2; Asn343 in S1 RDB. Protonation states of all the amino acid residues were predicted using PropKA version 3.0 at pH 7.0 [12]. The system was prepared for molecular dynamics using the AMBER18 software package and Leap AMBER toolkit, with the ff14SB force field [13]. Charges on the system were neutralized through the addition of counter-ions (Na^+^) and the system was placed in a with TIP3P water box with a minimum distance of 12 Å between the protein-surface and the side of the box, using the Leap module of AMBER.

The model was submitted to four consecutive minimizations stages to remove clashes prior to the MD simulation. In these four stages, the minimization procedure was applied to the following atoms of the system: 1-water molecules (2500 steps); 2-hydrogens atoms (2500 steps, divided in 1250 steps with the steepest descent algorithm and 1250 with the conjugated gradient algorithm); 2-hydrogens atoms (2500 steps, 1250/1250); 3-side chains of the amino acid residues (2500 steps, 1250/1250); 4-full system (10,000 steps, 5000/5000 steps). In stages 1, 2 and 3, the remaining atoms of the system were restrained to their initial position through a harmonic potential of 50 kcal mol^-1^ Å^-2^.

The model was then subjected to a molecular dynamics equilibration procedure, which was divided into two stages: in the first stage, the model was gradually heated to 310.15 K using a Langevin thermostat at constant volume (NVT ensemble) along 50 ps; in the second stage (additional 50 ps) the density of the system was further equilibrated at 310.15 K at constant volume. Finally, a molecular dynamics production run was performed for 400 ns with an NPT ensemble at constant temperature (310.15 K, Langevin thermostat) and pressure (1 bar, Berendsen barostat), using periodic boundary conditions, an integration time of 2.0 fs (with the SHAKE algorithm to constrain all covalent bonds involving hydrogen atoms). A 10 Å cutoff for nonbonded interactions was used during the entire molecular simulation procedure. Coordinates were saved at each 10 ps. Final trajectories were analyzed in terms of backbone root-mean-square deviation (RMSD), hydrogen bond formation and clusters analysis. The cluster analysis tool in cpptraj was used to partition the trajectory into 8 clusters, based on RMSD relative to the backbone C-alpha carbons, using the average-linkage hierarchical agglomerative method approach with a 2.0 Å cut-off. From the cluster analysis performed on the ensemble of poses generated from the MD simulations, one representative structure from each of the 8 dominant clusters was selected, for the subsequent step. The FPocket software (14) was then used to identify possible druggable pockets in the interfacial RBD-ACE2 region in each of the eight structures selected from the cluster analysis and in the initial model prepared from the X-ray structure. A minimum of 50 alpha-spheres per pocket was defined as criteria. 

### 2.2. Proteins and Antibodies

SARS-CoV-2 (438–516) S-RBD(HiS)_6_ including the minimal receptor binding motif previously described [11,14] and biotynilated human ACE2 have been purchased from Fisher Scientific (respective references 16534204 and 16545164). Monoclonal anti-6X His tag antibody has been purchased from ABCAM (reference ab18184, dilution 1/200). Anti α-tubulin antibody has been purchased from SIGMA (reference T6199, dilution 1/500). Secondary goat anti-mouse antibody coupled to AlexaFluor 488 was purchased from Fisher Scientific (Reference A11029, dilution 1/400). 

### 2.3. Pull Down Assay

Five hundred ng of each S-RBD(His)_6_ and ACE2-Biot were mixed for 15 min on ice in 10 µL of binding buffer 50 mM TrisHCl pH7.5, 0.1% Tween 20, 1 mM DTT, 10% glycerol and 150 mM NaCl. 12.5 µL of Dynabeads My One streptavidin T1 beads (Fisher Scientific, reference 10531265) were added to the solution and then incubated 30 min at 4 °C in rotation. Beads were then washed three times and proteins bound were eluted by incubation 3 min at 85 °C before loading on 12% sodium dodecyl sulfate-polyacrylamide gel electrophoresis (SDS-PAGE). Bound proteins were detected by western blot using anti-His antibody and streptavidin-HRP conjugate (SIGMA, reference GERPN1231).

### 2.4. AlphaLISA Binding Assays 

The AlphaLISA assay development was performed in 96-well ½ area Alpha plate (Perkin Elmer, Waltham, MA) with a final reaction volume of 40 µL. Cross titration experiments of ACE2-Biot (0.3 to 100 nM) against S-RBD(His)_6_ (0.3 to 100 nM) were carried out to establish optimal assay concentrations for the binding assay. Ten µL of each protein were diluted in the binding buffer 50 mM Tris-HCl pH 7.5, 150 mM NaCl, 0.1% (*v*/*v*) Tween-20, 0.1% (*w*/*v*) BSA, 1 mM dithiothreitol. The plate was incubated at room temperature (RT) for 2h in rotation. 10 µL of anti-6X His acceptor beads (Perkin Elmer, reference AL178) were then added and after 1 h of incubation at RT with rotation, 10 µL of streptavidin donor acceptor beads (Perkin Elmer, reference 6760002) were mixed to the wells. This established a final concentration of 20 µg/mL for each bead. The plate was then incubated for 1h at RT in the dark before the AlphaLISA signal was detected using an EnSpire Multimode Plate Reader (Perkin Elmer). Negative control with binding buffer was used to control the assay quality. Data were analyzed with GraphPad Prism 5.01 software. To evaluate tolerance for dimethyl sulfoxide (DMSO), the assay was performed as describe above with addition of 0.3 to 20% (*v*/*v*) of DMSO in the 3nM ACE2-Biot/S-RBD(His)_6_ binding step. In these conditions, binding data with DMSO were compared to the control condition without DMSO.

### 2.5. Bio-Layer Interferometry Binding Assays

Bio-Layer Interferometry (BLI) experiments were performed on a BLItz instrument (ForteBio) to measure the binding of S-RBD(His)_6_ to ACE2-Biot. All sample dilutions and baseline steps were carried out using the same reaction buffer (phosphate buffer saline pH 7.4, 0.02% (*v*/*v*) Tween-20, 0.1% (*w*/*v*) bovine serum albumin). Streptavidin biosensors (ForteBio, reference 18–5019) were first pre-wet for 10 min with the reaction buffer and 1 µM of ACE2-Biot was then loaded onto the coated biosensors for 500 s in order to reach a binding value of ~2.6 nm. Binding kinetics were divided in three steps. Firstly, baseline with the reaction buffer was measured for 60 s. For the association step, each loaded biosensor was dipped into serial dilutions of S-RBD(His)_6_ (from 3.12 nM to 100nM) for 300 s with a 2200 rpm shaking speed. Finally, the dissociation step of the bound S-RBD(His)_6_ was monitored by dipping the biosensor back into the reaction buffer for 300 s. Control experiments were performed to measure non-specific binding and included binding of reaction buffer with loaded biosensor (data subtracted) and binding of 100 nM of S-RBD(His)_6_ with no ACE2-Biot loaded onto the biosensor. The association and dissociation experimental curves were global fitted using a 1:1 binding model with the ForteBio Blitz pro 1.1 software. Kinetic binding parameters (k_a_, k_d_) and the equilibrium dissociation constant (K_D_) were determined. Sensorgrams curves were plotted using Prism 5.0 software (GraphPad Software, La Jolla, CA, USA)

### 2.6. Cells and Lentiviral Vectors Production

Lentivirus vector production was done by the service platform Vect’UB, (INSERM US 005—CNRS UMS 3427- TBM-Core, Université de Bordeaux, France). Lentiviral particles were produced by transient transfection of HEK293T (human embryonic kidney cells according to standard protocols. In brief, subconfluent HEK293T cells were co-transfected with lentiviral genome (psPAX2) (gift from Didier Trono (Addgene plasmid # 12260°), with an envelope coding plasmid (pMD2G-VSVG or wild type SARS-CoV-2 Spike protein (HDM_IDTSpike_fixK (kind gift from Bloom’s laboratory, (16)) and with vector constructs (44°:pRRLSIN-PPT-hPGK-eGFP-WPRE or pHAGE_EF1alnt_ACE2_WT) by calcium phosphate precipitation. LVs were harvested 48h post-transfection and concentrated by ultracentrifugation. Viral titers of VSV-g pseudotype pLV lentivectors were determined by transducing HEK293T cells with serial dilutions of viral supernatant and eGFP expression was quantified five days later by flow cytometry analysis. For SARS-CoV-2 Spike pseudotype, p24 antigen levels were measured in the concentrated viral supernatants by an enzyme-linked immunosorbent assay (Innotest HIV Ag nAb; Fugibio, France) and viral titers were estimated by comparing p24 antigen levels of each lentiviral supernatant with a similar VSV-g pseudotype lentiviral supernatant produced simultaneously.

HEK293T-ACE2 cell lines have been generated by lentiviral transduction using pHAGE_EF1alnt_ACE2_WT plasmid (kind gift from Bloom’s laboratory [15]). HEK293T cells (200,000 cells) were then transduced with optimized concentration of ACE2 lentiviral particles in 6-well plates. The efficacy of transduction was assessed using real time PCR ten days post infection. ACE2 provirus DNA from this cell line was quantified by q-PCR using the ΔCt method as compared with 1 copy cell line. DNA from two different cell clones (human 293T and K562 cells), containing a single integrated copy of the provirus, was used as a normalized cell line. HEK293T were cultured in DMEM medium supplemented with gentamycin 5 mg/mL and FBS (Fetal Bovine Serum) 10%. 

### 2.7. Quantitative PCR of Proviral DNA

DNA was extracted from 1 × 10^6^ cells with NucleoSpin Tissue kit (Macherey-Nagel) and real-time PCR was performed on 30 ng of DNA with SYBR Green kit GoTaq^®^ qPCR Master Mix (*Promega).* DNA from non-transduced cell was isolated and amplified at the same time to demonstrate the absence of contamination. Samples were heated to 50 °C for 2 min and to 95 °C for 10 min and then 40 cycles of PCR were performed with 15 s at 95 °C and 1 min at 60 °C. The following RRE sequences were used: 5′-GTGCAGCAGCAGAACAATTT-3′ and 5′-GATGCCCCAGACTGTGAGTT-3′ as primers. DNA from a HEK cell line with one proviral insert (HEK-2C9) was used as a standard for quantification by the ΔCt method.

### 2.8. Cell Imaging

For cell imaging 20,000 HEK293T and HEK293T-ACE2 cells were plated on glass coverslips pretreated with poly-L-Lysine solution 0.01% (SIGMA ref P4832) 5 min at room temperature. After viral transduction cells were fixed by 5 min PFA 4% treatment and washed two times with PBS. Transduction and cell-cell fusion were checked over time directly using EVOS device imaging before PFA fixation. For monitoring and quantification of cell-cell fusion the cells were DAPI stained (DAPI SIGMA Ref D9542 100ng/mL in PBS) and imaging was performed using inverted epifluorescence microscopy and image acquisition (phase and DAPI channels). Epifluorescence microscopy was carried out on a Zeiss Axioimager Z1 driven by Metamorph. 

For immunofluorescence cells were first permeabilized by treatment 15 min with triton X-100 0.1%, saponine 0.4% in PBS then saturated for 1 h with BSA 2%, saponine 0.1% in PBS. Cells were then incubated with the primary antibody for 1 h in the saturating buffer followed with three washes with PBS saponine 0.1%. The cells were then incubated with the secondary antibody for 45 min in the saturating buffer. After three washes with the saturating buffer the cells were treated 10 min with DAPI in PBS. After two PBS washes and one final H_2_O wash the coverslip were sealed with 3 µL of Prolong Diamond Antifade reagent (Fisher Scientific Ref P36965). 

### 2.9. In Cellulo Imaging of the SARS-CoV-2 S-RBD/ACE2 Interaction

10,000 HEK293T and HEK293T-ACE2 plated on glass coverslips pretreated with poly-L-Lysine solution 0.01% (SIGMA ref P4832) 5 min at room temperature. After 24 h. the RBD recombinant protein (4–40 pmoles) was added to the cells in 100 µL PBS. Cells were then washed and fixed (PFA 4%) at different time points. Cell imaging was performed as described above. 

For quantification of the S-RBD/ACE2 interaction in cellular context 45,000 HEK293T and HEK293T-ACE2 cells were incubated 45 min at 37 °C in 100 µL Dulbecco’s Modified Eagle’s Medium (DMEM) and increasing concentrations of RBD recombinant protein. One ml of PBS was added the cells were centrifugated 5 min at 2500 rpm. 50 µL of anti-His antibody 1/200 in DMEM was added and the cells were further incubated 45 min at 37 °C. After PBS washes 50 µL of a 1/400 DMEM solution of secondary antibody was added and cells were incubated 30 min,s at 37 °C. After PBS washes the cells were resuspended in 200 µL PBS, FBS 2% EDTA 2mM and the percentage of FITC positive cells was quantified by flow cytometry. 

## 3. Results

### 3.1. In Silico Molecular Dynamics Simulation of the SARS-CoV-2 S/ACE2 Complex

To monitor the S/ACE2 complex and provide data about the dynamics of the interaction, we used previously published structural data [11] for 400 ns molecular dynamics simulations with the AMBER force field and TIP3P water box. Simulations were run with 0.150 M NaCl in a neutral simulation box subject to periodic boundary conditions, in line with previous molecular dynamics studies on the SARS-COV-2 S1 RDB complexed with ACE2 [16,17,18]. The simulation allowed us to precisely model the structural interfaces within the complex (Figure 1A) and isolate the receptor binding domain of S (RBD). To precisely monitor the contacts between the RBD spike protein and ACE2, we considered the previously published X-ray structure [11] and 8 representative structures obtained from a cluster analysis performed on a 400 ns molecular dynamics simulation with the AMBER force field (ff14SB) in a TIP3P water box with periodic boundary conditions (1 representative structure per cluster). These 9 structures were used with FPocket [19] to search for and identify important zones of contact between the two proteins within the S/ACE2 complex. The protocol allowed us to identify transient interfaces within the S/ACE2 complex and to incorporate the dynamic nature of the interface into the definition of the pocket to be explored in future virtual screening studies. The structural pockets identified are given in Figure 1B, which also illustrates the representative structure of each cluster and the percentage of structures in the MD simulations that correspond to each cluster. These pockets are further characterized in Table 1, in terms of total solvent accessible surface area (SASA), polar SASA, non-polar SASA, volume and main contributing amino acid residues at the ACE2 and S1 RDB regions. In addition, from the analysis of the hydrogen bond patterns during the MD simulations, the most stable hydrogen interactions for the S/ACE2 association were identified. These are reported in Figure 1C, which reports all the interfacial hydrogen bonds that are present in more than 20% of the structures generated by MD. This suggests that targeting the pockets identified in Figure 1B in a way that interferes with or blocks the formation of the hydrogen bonds described in Figure 1C may constitute a strategy to be used in virtual screens for new compounds.

### 3.2. In Vitro Monitoring of the SARS-CoV-2 S-RBD/ACE2 Complex and Determination of Kinetic Constants

To directly monitor the interaction between the minimal ACE2 interacting domain of the SARS-CoV-2 S protein, we first performed pulldown assays using the minimal RBD domain defined in the computational modeling. For this purpose, the recombinant (438–516) RBD protein including the minimal receptor binding motif RBM (438–506) that interacts directly with ACE2 fused to a 6xhistidine tag [11,14] and the full-length human ACE2 protein fused to a biotin group were subjected to pulldown assays. As shown in Figure 2A, the two proteins were found to bind to one another specifically in pulldown assays performed using streptavidin-coupled beads. The optimal interaction conditions were established in 50 mM Tris-HCl pH 7.5, 0.1% Tween 20, 1 mM DTT, 10% glycerol and 150 mM NaCl. A total of 500 ng of each protein was sufficient to detect both proteins after SDS-PAGE separation and western blot with anti-HIS antibody and streptavidin coupled to HRP. NaCl concentrations up to 350 mM did not significantly change the binding efficiency (Figure 2B), indicating that the interaction was stable.

These optimal binding conditions were then used to set up AlphaLISA monitoring of the RBD/ACE2 interaction. The AlphaLISA assay was adapted to 96-well plates in a final reaction volume of 40 µL and 2 h of incubation. Anti-6XHis acceptor and streptavidin donor beads were used for complex capture after 2 h of incubation of the two partners. Cross titration experiments of ACE2-Biot (0.3 to 100 nM) against S-RBD(His)_6_ (0.3 to 100 nM) were carried out to establish optimal assay concentrations for the binding assay. As reported by the cross-titration experiments shown in Figure 2C, typical Gaussian interaction curves were obtained with a maximal binding signal of up to 1 million AS units using 0.3–100 nM protein concentration. More accurate titration showed that a strong AS signal could be obtained using low protein concentrations between 0.625–10 nM (Appendix A). From the perspective of testing potential drugs, we also analyzed the tolerance of the interaction to the presence of DMSO solvent. As reported in Figure 2D, DMSO concentrations up to 5% did not significantly affect the interaction signal, which appears suitable for main drug screens.

To monitor the dynamics of the RBD/ACE2 interaction and measure the binding parameters, we applied biolayer interferometry (BLI) to the complex. First, the binding of the RBD protein on the corresponding biosensor with increasing concentrations of protein was monitored (Appendix A). After determination of the optimal ACE2 binding to the biosensor, we applied increasing concentrations of S-RBD(His)_6_ as described in the Materials and Methods section. Binding kinetics were divided into three steps: baseline with reaction buffer measurement for 60 s and association and dissociation steps for both proteins. Control experiments were performed to measure nonspecific binding and included the binding of reaction buffer with a loaded biosensor (data subtracted) and the binding of 100 nM of S-RBD(His)_6_ with no ACE2-Biot loaded onto the biosensor. As reported in Figure 3A, strong binding of S-RBD(His)_6_ to ACE2 was observed. The kinetic binding parameters (k_a_, k_d_) and the equilibrium dissociation constant (K_D_) were determined after global fitting of the association and dissociation curves using a 1:1 binding model with ForteBio Blitz pro 1.1 software. The K_D_ was found to be approximately 0.93 nM, as reported in Figure 3B, confirming previous data [3].

### 3.3. In Cellulo Monitoring of S/ACE2-Mediated Viral Entry

To monitor the S/ACE2 complex in a more physiological context and thus study the viral entry process, we set up a lentiviral vector-based cellular approach. For this purpose, lentiviral vectors pseudotyped with the wild type SARS-CoV-2 spike protein (CoV-2 LV) were produced from HEK293T cells transfected with three plasmids: a plasmid encoding a lentiviral backbone (LV44: pRRLSIN-PPT-hPGK-eGFP-WPRE) expressing a marker protein, the HDM_IDTSpike_fixK plasmid expressing the Spike protein (kind gift from the Bloom laboratory [15]), and the psPAX2 plasmid expressing the other HIV proteins needed for virion formation (Tat, Gag-Pol and Rev). HEK293T-ACE2 cells were generated by lentiviral vector insertion of the human *ACE2* gene in their genome. Selection of a cell line carrying 9 copies of the *ACE2* gene, quantified by qPCR, was performed (Appendix A). The CoV-2 LV was shown to transduce only HEK293T-ACE2 cells and not wild-type HEK293T cells, leading to the integration and expression of the *eGFP* gene encoded by the LV (Figure 4A). This confirmed that the transduction efficiency indicated by the percentage of eGFP-positive cells relies on the S/ACE2 interaction. In contrast, typical VSVg-pseudotyped LVs were able to transduce both 293T and 293T-ACE2 cells.

Soluble ACE2 recombinant protein has been shown to block SARS-CoV-2 infection in cells [20]. To further validate our system and better confirm the S/ACE2-dependent viral entry of the CoV-2 LV, we performed competition/neutralization experiments using soluble ACE2 protein. As reported in Figure 4B, adding increasing concentrations of ACE2 led to a strong inhibition of CoV-2 LV infectivity, in contrast to VSVg LV, which was not susceptible to ACE2 treatment. Treatment with drugs targeting the early stages of lentiviral replication, that is, reverse transcription and integration, which are shared in common by the two LV systems (efavirenz treatment reported in Figure 4C and dolutegravir treatment reported in Figure 4D), also confirmed that both LVs were equally affected by the drugs. This confirmed that CoV-2 pseudotyping did not affect the further early steps of LV infection, allowing, by comparing the two systems, the specific selection of conditions or drugs that specifically affect the CoV-2 entry step. Taken together, these data fully confirmed that LV infectivity monitoring of CoV-2 was directly related to the S/ACE2 entry pathway. This result also confirmed that our COV-2 LV infectivity assay could serve as a basis for selecting drugs specifically targeting the entry of this virus.

### 3.4. In Cellulo Monitoring of the S/ACE2-Mediated Cell-Cell Fusion

The viral S/cellular ACE2 interaction has been shown to be responsible for the cell-cell fusion induced by SARS-CoV-2 infection [21]; thus, we wondered whether CoV-2 LV transduction might also induce these cell fusions in our infectivity assay. After transduction with CoV-2, LV HEK293T-ACE2 cells were observed at different time points under a light microscope to visualize possible cell fusions. As reported in Figure 5A and magnified in Figure 5B, typical cells with multiple nuclei were detected after transduction with the CoV-2 vector. This was further confirmed by tubulin immunostaining showing polynuclear cells with increased size (Figure 5C). In contrast, no cell fusion could be observed in non-transduced cells or in cells transduced with VSVg LV. Analysis of the kinetics of cell fusion showed that the maximal percentage of fused cells was observed from 2 h post-transduction and lasted until 16 h post-transduction (Figure 5D).

To further confirm that the cell fusion phenotype was due to the S-ACE2 interaction, we performed ACE2 neutralization experiments by adding soluble protein as a competitor for S binding. As reported in Figure 5D, the triggering of cell fusion was dramatically impaired when ACE2 was added to the medium. In contrast, treatment with inhibitors of early lentiviral replication events, such as efavirenz or dolutegravir, did not change the cell fusion proportion, though they did inhibit LV infectivity, as expected from their effect on the intracellular reverse transcription and integration steps, respectively.

Since cell fusion is related to the interaction between S and ACE2, we next wondered whether the S-RBD protein could directly trigger fusion. To answer this question, we incubated either HEK293T or HEK293T-ACE2 cells with the recombinant S-RBD protein and monitored cell fusion using epifluorescence microscopy to quantify polynuclear cells. As reported in Figure 6A, incubation of HEK293T-ACE2 cells with S-RBD protein induced a strong cell fusion phenotype. In contrast, no detectable effect was observed in HEK293T cells lacking the ACE2 receptor. The morphology of fused cells, magnified in Figure 6B, was similar to that observed in 293T-ACE2 cells transduced with the CoV-2 LV (cf. Figure 5B). Quantification of the percentage of polynuclear cells confirmed the specific effect of the S-RBD protein in HEK293T-ACE2 cells compared to HEK293T cells that did not express the ACE2 receptor (Figure 6C). Treatment with recombinant soluble ACE2 protein confirmed that the cell fusion induced by S-RBD protein could also be prevented by competition with the cellular soluble receptor in contrast to efavirenz or dolutegravir (Figure 6D).

All these data paralleled the infectivity data and fully confirmed that SARS-CoV-2 LV infectivity in HEK293T-ACE2 cells was related to the S/ACE2 interaction and that the cell fusion phenotype could be an additional readout of this association. Furthermore, our data also indicated that S-RBD was minimally required for triggering cell fusion.

### 3.5. In Cellulo Monitoring and Quantification of the S/ACE2 Interaction

S/ACE2 interaction during infection occurs in the cellular membranous environment that is not recapitulated in the *in vitro* interaction systems. In order to directly monitor the S-RBD/ACE2 interaction under cellular conditions and provide a method to quantify this cellular association we developed an immunofluorescence binding assay using recombinant RBD(His)_6_ previously used for the *in vitro* interaction assays. For this purpose, recombinant S-RBD(His)_6_ protein was incubated with either HEK293T-ACE2-expressing cellular receptors or control HEK293T cells. Cells were then observed under an epifluorescence microscope after immunofluorescence labeling using anti-(His)_6_ antibodies. As reported in Figure 7A,B, after 16 h incubation with 293T-ACE2 cells, the S-RBD(His)_6_ protein was detected mostly at the plasma membrane. In contrast, no significant signal could be detected when the protein was incubated with 293T cells lacking the ACE2 receptor. More accurate kinetic analysis of incubation showed that the S-RBD protein was detected 2 h after incubation and the plasma membrane signal increased with time, reaching a maximum intensity at 16 h (Figure 7C). To better quantify RBD binding to the cellular ACE2 receptor, we measured the percentage of cells that were positive for RBD binding by anti-(His)_6_ immunofluorescence followed by flow cytometry quantification. As reported in Figure 7D, RBD binding to HEK293T-ACE2 cells could be detected using anti-(His)_6_ primary antibody and Alexa488-coupled secondary antibody. The positive signal was found to be specific for the presence of the cellular ACE2 receptor, confirming that the RBD-ACE2 interaction could be monitored and quantified. ACE2 competition assays further verified these data, as shown by the RBD binding to the cellular ACE2 receptor when soluble protein was added to the assay (Figure 7E).

## 4. Discussion

SARS-CoV-2 infection relies on the early interaction between the viral spike protein S and its cellular receptor ACE2, which triggers the entry of the virus into host cells. The fundamental understanding of this entry step and the subsequent development of specific and efficient inhibitors of this crucial replication stage, requires the ability to monitor the S/ACE2 interaction *in silico, in vitro* and *in cellulo and* this specific interaction remains difficult to isolate in the context of a full wt virus infection. All three monitoring approaches were developed in combination in this work to help further analyze S-ACE2-mediated viral entry processes.

Based on previously published structural data [11], 400 ns molecular dynamics simulations were performed, providing a dynamic view of the interaction between the RBD of the S protein and ACE2 that complements the information provided from the available X-ray structures. Together, these results provide detailed atom-level data on the minimal regions of interaction, which suggest possible druggable sites. In addition to providing data about the dynamics of the interaction, this work allowed us to set up modeling pipelines for further virtual screening of drug libraries targeting these unedited druggable sites. A set of biochemical and biophysical parameters was then used to both monitor and characterize this interaction between the minimal S-RBD/ACE2 complex. In particular, AlphaLISA technology could be applied to detect the interaction between both proteins. Optimization of this approach allowed us to monitor the association between the two partners with very low amounts of proteins (nM range) and strong AS signals (up to 1,000,000 AS units). Tolerance to DMSO solvent was also analyzed to facilitate further drug analysis. This powerful system will allow drug library screening as well as characterization of the interaction in different genetic contexts, such as S mutants from currently circulating SARS-CoV-2 variants. An additional optical analytical technique based on biolayer interferometry (BLI) was also used to detect the S-RBD/ACE2 interaction. In addition to providing monitoring of the interaction in a dynamic context, the BLI data analysis allowed us to measure a fast association constant (k_a_ = 2.65 × 10^5^ M^−1^s^−1^) and slow dissociation constant (k_d_ = 2.47 × 10^−4^ s^−1^). The equilibrium dissociation constant (K_D_) was 0.93 nM. All these results indicate a very stable and strong interaction between the S-RBD and ACE2 proteins. This assay will be very useful for testing the influence of S or ACE2 mutants, drugs and additional cofactors on both the association and dissociation steps of the S/ACE2 interaction process.

In addition to these biochemical approaches, we developed cellular systems as complementary models to monitor the S/ACE2 interaction in a physiological context. An infectivity assay relying on the S/ACE2 interaction was set up using lentiviral vectors pseudotyped with SARS-CoV-2 S protein and HEK293T cells expressing the human ACE2 protein. A comparison with the infectivity of lentiviruses pseudotyped with the VSVg envelope protein showed that the SARS-CoV-2 LV infectivity relied on typical S/ACE2 interaction suggesting that the main mechanism of viral entry as S structural rearrangements occurring during viral infection also take place in our system. These results enable us to study parameters specifically modulating the SARS-CoV-2 entry step. This system will allow us to determine, or screen, drug effects on this specific S/ACE2-dependent viral entry. Indeed, adding increasing concentrations of soluble ACE2 protein led to a strong inhibition of CoV-2 LV infectivity, in contrast to VSVg LV, which was not susceptible to ACE2 treatment. This confirmed that CoV-2 LV infectivity monitoring was directly related to the S/ACE2 entry pathway and that this assay could serve as the basis for selecting drugs specifically targeting this viral entry. Interestingly, transduction with the SARS-CoV-2 LV vectors, in contrast to VSVg LV, induced a cell fusion phenotype dependent specifically on the S/ACE2 interaction and thus recapitulated the cell fusion observed with wt virus [21]. Additionally, adding the recombinant RBD protein to cells expressing ACE2 and not the parental cells directly triggered cell fusion, suggesting that the minimal S-RBD domain was sufficient for inducing this process. The S-RBD/ACE2 interaction could be monitored both by immunofluorescence and by FACS analysis using HEK293T-ACE2 cells and S-RBD(His)_6_ protein. In addition to providing an additional assay for directly detecting the RBD/ACE2 interaction in a physiological context, this model would also allow us to precisely determine the mechanism of action of drugs targeting the S/ACE2 complex as well as investigate this specific event. 

In summary, we provide here a panel of *in silico*, biochemical, biophysical and cellular approaches for monitoring the SARS-CoV-2 S/human ACE2 interaction. This set of approaches is available for the community to further investigate the viral entry pathway and characterize new drugs or viral variants. The study reported here has been focused on wild type S protein study while additional variants are currently spreading worldwide including the D614G dominant SARS-CoV2 species. The models reported here may provide additional tools to characterize specifically the cellular entry efficiency of these variants. These approaches will also specifically allow the characterization of factors modulating the S/ACE2 complex and the study of their effects on the direct interaction between viral and cellular partners or on the cell fusion or infectivity induced by this association. Indeed, multiple parameters have been reported to modulate the S/ACE2 mediated viral entry as S stability, conformational changes occurring during the fusion process and glycosylation [16,22,23]. Furthermore, cellular factors incorporated in the SARS-CoV-2 LV used in our study may also participate in cell entry and/or cell fusion as well as lipid composition of the LV membrane. The reported methods could thus also be used for a better understanding of the role of these processes in S functions and for investigating additional determinants of these mechanisms. Finally, *in silico* molecular dynamics simulation of the SARS-CoV-2 S/ACE2 complex revealed structural pockets within the S/ACE2 complex; targeting this structure in such a way that blocks the pocket or interferes with the formation of the described hydrogen bonds may be a strategy for future virtual screens for new compounds. This may constitute complementary approaches to previous studies aiming to search for drugs targeting the S/ACE2 interfaces [24]. All approaches reported in this work would now allow the molecular and cellular analyzes of the SARS-CoV-2 entry step.

## Figures and Tables

**Figure 1 viruses-13-00365-f001:**
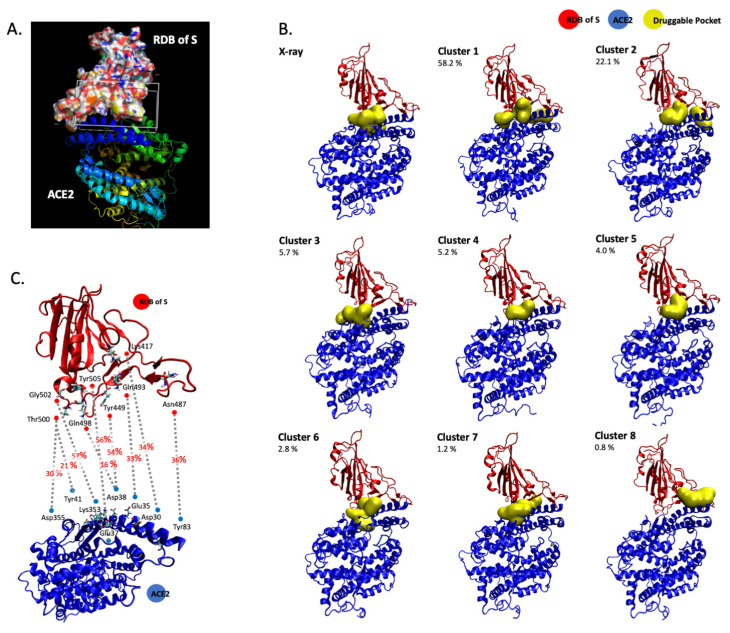
Computational modelling of the severe acute respiratory syndrome coronavirus 2 (SARS-CoV-2)/ACE2 complex. The S RDB-ACE2 complex has been modelled using molecular dynamics calculations (**A**). In the image the surface representation represents the RDB of the S protein, while the ribbon representations describes ACE2. Druggable binding pockets at the RDB-ACE2 interface identified with FPocket in the X-ray structure (6M0J) and in the 8 representative structures obtained from the cluster analysis performed on the 400 ns MD simulation are reported in (**B**). The values reported refer to the percentage of structures within each cluster in the MD simulation. Dominant hydrogen bonds and participating residues at the RDB/ACE2 interface with indication of the percentage of MD structures in which they are formed are reported in (**C**).

**Figure 2 viruses-13-00365-f002:**
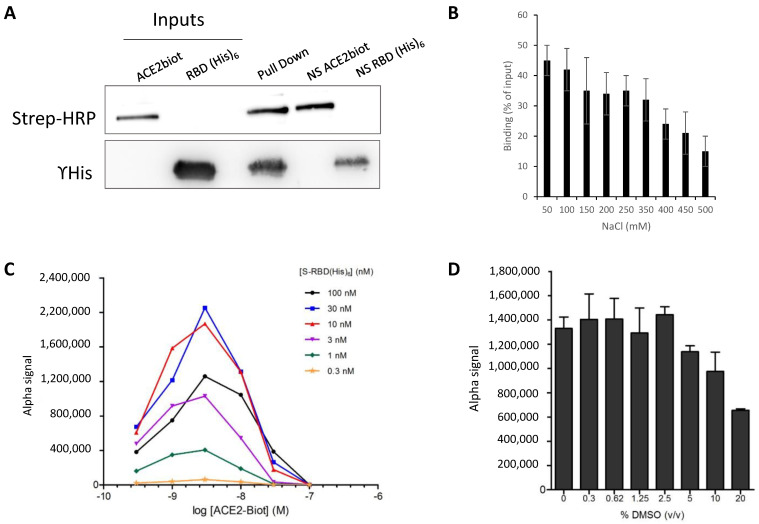
*In vitro* monitoring of the SARS-CoV-2 S-RBD/ACE2 interaction. SARS-CoV-2 S-RBD(His)_6_ and human ACE2-Biot interaction have been first monitored by typical pull down assay using streptavidin coupled magnetic beads (**A**) using 500 ng of each protein. Input, co-precipitated proteins (pull down) and non-specific binding of each partner on beads are reported in the figure. Pull downs have been performed at different NaCl concentration and the percentage of retained S-RBD protein is reported in (**B**). Data were obtained from at least three independent experiments and are reported as means ±SD. The S-RBD/ACE2 complex has been further analyzed using AlphaLISA assay. Cross titration experiments of ACE2-Biot (0.3 to 100 nM) against S-RBD(His)_6_ (0.3 to 100 nM) using anti-6X His acceptor beads and streptavidin donor acceptor beads are reported in (**C**). Evaluation of the tolerance for dimethyl sulfoxide (DMSO) has been performed as describe above with addition of 0.3 to 20% (*v*/*v*) of DMSO in the ACE2-Biot/S-RBD(His)_6_ binding step. In these conditions, binding data with DMSO were compared to the control condition without DMSO (**D**). Data were obtained from at least two independent experiments and are reported as means ±SD.

**Figure 3 viruses-13-00365-f003:**
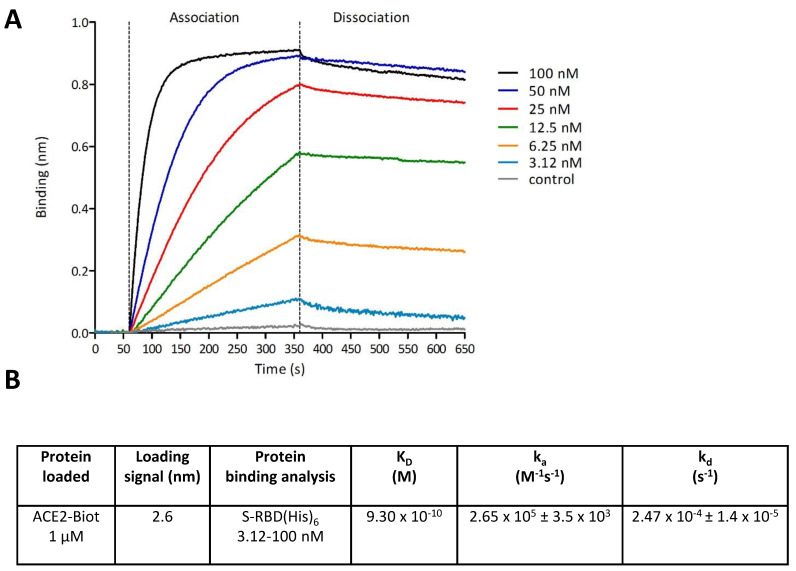
Monitoring of the SARS-CoV-2 S-RBD/ACE2 interaction using Bio-Layer Interferometry. Bio-Layer Interferometry (BLI) experiments were performed using Streptavidin biosensors and 1 µM of ACE2-Biot coated for 500 s in order to reach a binding value of ~2.7 nm. Baseline with the reaction buffer was measured for 60 s. For the association step, each loaded biosensor was dipped into serial dilutions of S-RBD(His)_6_ (from 3.12 nM to 100 nM) for 300 s. The dissociation step of the bound S-RBD(His)_6_ was monitored by dipping the biosensor back into the reaction buffer for 300 s. Control experiments were performed to measure non-specific binding and included binding of reaction buffer with loaded biosensor (data subtracted) and binding of 100 nM of S-RBD(His)_6_ with no ACE2-Biot loaded onto the biosensor. Sensorgrams curves shown in (**A**) were plotted using Prism 5.0 software (Graphpad Software, La Jolla, CA). The association and dissociation experimental curves were global fitted using a 1:1 binding model with the ForteBio Blitz pro 1.1 software. Kinetic binding parameters (k_a_, k_d_) and the equilibrium dissociation constant (K_D_) were determined and reported in (**B**).

**Figure 4 viruses-13-00365-f004:**
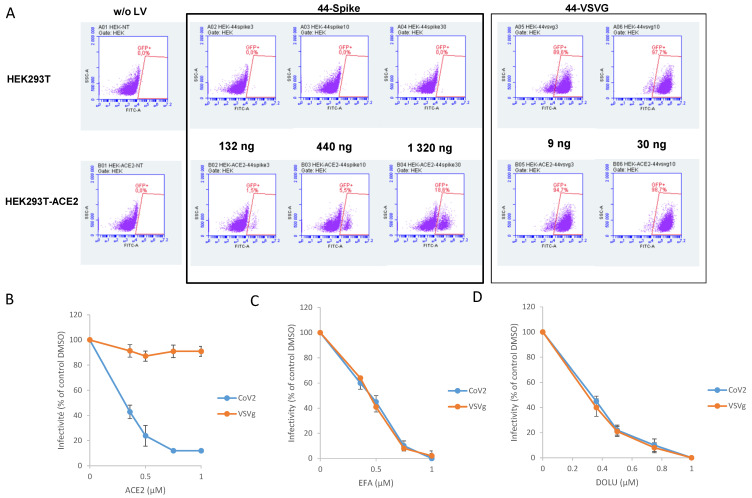
VSVg and SARS-CoV-2 pseudotyped lentiviruses infectivity assay. HEK293T and HEK293T-ACE2, generated as described in the materials and methods section, were transduced with lentiviral vectors peseudotyped with SARS-CoV-2 S protein (44-spike) or VSVg protein (44-VSVg) using different quantities of viral particles reported in the figure as ng of p24. Infectivity has been evaluated by eGFP-positive cells quantification by flow cytometry (**A**). Competition experiments have been performed using soluble ACE2 protein added simultaneously to the lentiviral vectors (**B**). Efavirenz, inhibitor of retroviral reverse transcription steps, and Dolutegravir, inhibitor of the integration step of retroviral replication, have been added simultaneously to VSVg and CoV-2 lentiviral vectors (respectively **C**,**D**). Data were obtained from at least three independent experiments and are reported as means ±SD.

**Figure 5 viruses-13-00365-f005:**
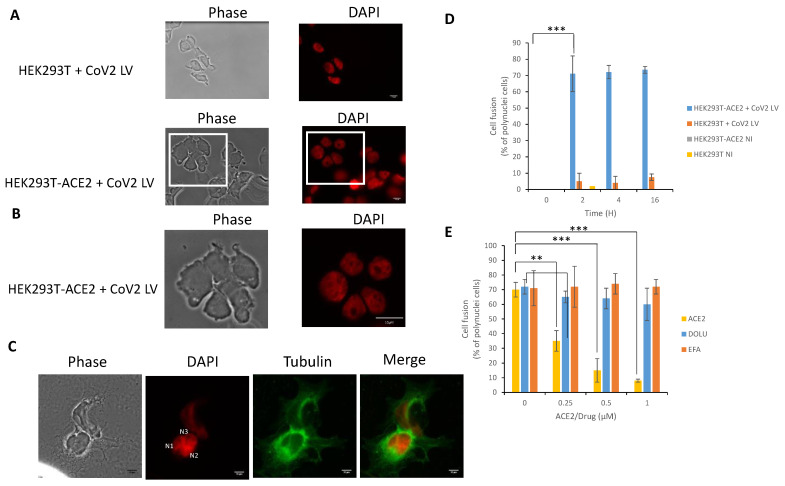
Cell-cell fusion induced by SARS-CoV-2 LV transduction. After transduction of HEK293T and HEK293T-ACE2 with CoV-2 LV, cells were DAPI stained and visualized by epifluorescence microscopy 0–16 h post-transduction (16 h’ time point monitoring in **A**, magnification in **B**). Cell-Cell fusions have been monitored by immunofluorescence using anti-α-tubulin and DAPI staining (**C**). The number of cell fusion has been quantified by measuring the number of polynuclei cells (**D**). NI: non infected. The effect of soluble ACE2, Efavirenz and Dolutegravir has been analyzed by adding the protein, or drugs, simultaneously with the LVs and by quantifying the cell fusion 16 h post-transduction (**E**). Data were obtained from at least two independent experiments and are reported as means ±SD. A least 50 cells have been quantified in each conditions. ** *p* < 0.05; *** *p* < 0.001.

**Figure 6 viruses-13-00365-f006:**
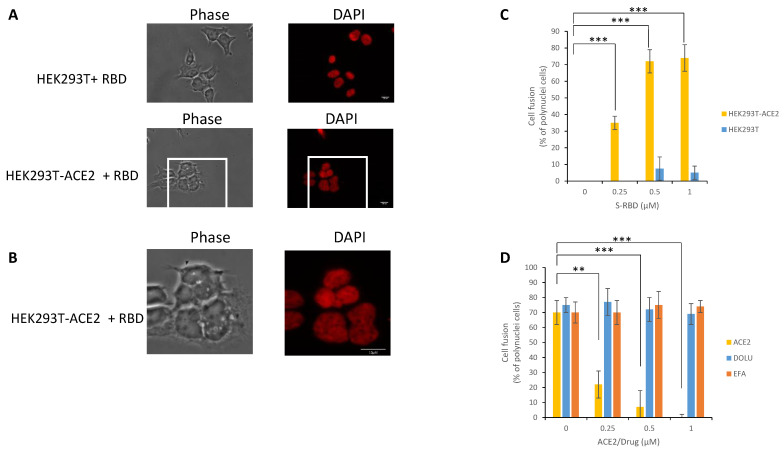
S-RBD meditated cell fusion. Increasing concentrations of SARS-CoV-2 S-RBD fragment were added to either HEK293T or HEK293-ACE2 cells for 0–16 h before epifluorescence microscopy (1 µM RBD and 16 h’ time point in (**A**) and magnification in (**B**)). The number of cell fusions observed after incubation with increasing concentration of RBD has been quantified by measuring the number of polynuclei cells (**C**). the effect of soluble ACE2, Efavirenz and Dolutegravir has been analyzed by adding the protein or drugs simultaneously with the RBD and by quantifying the cell fusion 16 h after treatment (**D**). Data were obtained from at least two independent experiments and are reported as means ±SD. At least 50 cells have been quantified in each conditions. ** *p* < 0.05; *** *p* < 0.001.

**Figure 7 viruses-13-00365-f007:**
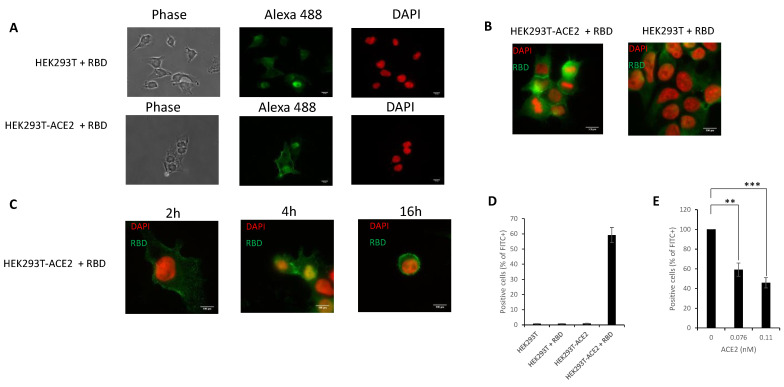
Cell monitoring and quantification of the S-RBD/ACE2 interaction. The SARS-CoV-2 S-RBD fragment was added to either HEK293T or HEK293-ACE2 cells for 16 h and immunofluorescence has been performed using anti-(His)_6_ antibody and secondary antibody coupled to Alexa-Fluo 488. Cells were observed by epifluorescence microscopy (**A**). A typical DAPI/Alexa 488 channels merge is reported in (**B**) and RBD distribution profile in HEK293T-ACE2 cells over time is reported in (**C**). Cells were also analyzed using flow cytometry to detect the percentage of cells positive for FITC signal in the absence (**D**) or in the presence of increasing concentration of ACE2 soluble protein (**E**). ** *p* < 0.05; *** *p* < 0.001.

**Table 1 viruses-13-00365-t001:** Characterization of the binding pockets identified on the X-ray structure and on the 8 representative structures obtained from the cluster analysis in terms of SASA (solvent accessible surface area), volume and main interacting amino acid residues at the angiotensin-converting enzyme 2 (ACE2) and S1 RBD interface. Values obtained with FPocket.

Cluster	Pocket	Total SASA (Å^2^)	Polar SASA (Å^2^)	Non-Polar SASA (Å^2^)	Volume (Å^3^)	ACE2 Main Residues	S1 RDB Main Residues
X-ray	1	141.0	75.8	65.2	351.1	Asn33, His34, Glu37, Lys353	Arg403, Lys417, Tyr453, Tyr495, Tyr505,
	2	193.8	98.4	95.4	738.9	Gln42, Leu45, Asn49, Asn61, Asn64, Lys68	Gly446, Tyr449
	3	241.4	115.8	125.6	827.5	Thr324, Phe356, Arg393	Gly502, Val503, Tyr505
Cluster 1	1	223.5	134.1	89.4	828.7	Asn33, His34, Glu37, Asp38, Lys353, Pro389, Phe390	Arg403, Glu406, Lys417, Ile418, Tyr495, Tyr505
	2	155.7	79.6	76.1	543.4	Phe28, Lys31, Phe32, Glu35, Asp38, Leu39, Phe72, Gln76, Leu79	Glu484, Phe486, Asn487, Tyr489
	3	152.0	43.3	108.7	525.8	Thr324, Gln325, Asn330, Asp355, Phe356	Val503, Tyr505
Cluster 2	1	304.7	173.1	131.6	919.7	His34, Glu37, Asp38, Lys353, Pro389, Arg393	Arg403, Gln409, Lys417, Ile418, Tyr495
	2	174.4	112.2	62.2	549.8	Lys31, Gln76, Leu79	Glu484, Tyr489, Phe490
Cluster 3	1	207.3	98.6	108.7	603.1	Lys353, Phe356, Arg393	Asp405, Tyr505
	2	257.4	142.6	114.7	750.6	Thr324, Gln325, Glu329, Asp355	Thr500, Gly502, Val503
Cluster 4	1	213.1	128.6	84.5	642.8	Asn33, His34, Pro389, Arg393	Arg403, Asp405, Lys417
Cluster 5	1	232.1	122.2	109.9	784.6	Asn33, His34, Pro389, Phe390	Asp405, Lys417, Ile418, Tyr495, Tyr505
Cluster 6	1	155.8	96.6	59.2	535.4	Asp30, Asn33, His34, Glu37, Asp38, Lys353, Phe390	Arg403, Lys417, Tyr495, Tyr505
	2	187.7	75.4	112.3	566.0	Asp350, Phe356, Tyr385, Arg393	Gly502, Tyr505
Cluster 7	1	175.3	107.7	67.6	565.9	Glu37, Asp38, Leu39, Lys353	Leu452, Arg403, Gln493, Tyr495, Tyr505
	1	285.5	128.5	157.0	833.7	Asn33, Thr324, Gln325, Phe356, Pro389, Arg393	Arg403, Asp405, Val503
Cluster 8	1	252.0	144.5	107.5	799.2	Thr20, Glu23, Gln24, Lys26, Thr27, Asp30	Lys417, Tyr421, Phe456, Tyr473, Ala475

## Data Availability

The data presented in this study are available on request from the corresponding author.

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
