# Peer review of "In Silico*, *In Vitro* and *In Cellulo* Models for Monitoring SARS-CoV-2 Spike/Human ACE2 Complex, Viral Entry and Cell Fusion"

_viruses, 2021, doi:10.3390/v13030365_

Round 1
Reviewer 1 Report
Lapaillerie and colleagues describe in this manuscript a combination of in silico, in vitro and tissue culture based approaches for the investigation of interaction of the novel SARS-CoV-2 virus with target cells via the viral spike protein and human ACE2. The authors generated a simulation model of the using published X-ray crystallographically determined structures of the viral spike protein, including the RBD, and ACE2. The simulations resulted in six clusters proposed to be important for the interaction and potentially useful as drug targets. The in vitro and tissue culture approaches employ molecular biological approaches. The isolated proteins or RBD are used to assess in vitro interactions under conditions that may allow for testing of potential drugs in the presence of chemicals, such as DMSO, that increase solubility of lipohilic compounds. The authors also generated a lentiviral constructs (LV) to deliver proteins into cells and a LV pseudotyped with SARS-CoV-2 spike protein and GFP that allows tracking the infection. The study is overall straight forward and the described tools may provide very useful for future studies aiming at disruption of the SARS-CoV-2 with human target cells. There are however a few questions the authors may want to clarify. The justification for some of the conditions / assumptions used for the simulations is not clear. Ionic strength, charge and solubility near membranes can have a decisive effect on protein-protein interactions, in particular those that result in membrane fusion. Related to that point, a discussion for the parameters that may be important for the in vitro interaction assay will be useful. How comparable are the surfaces used for the in vitro assays with the environment near a membrane where the viral spike protein and ACE2 interact? The LV pseudotyped with SARS-CoV-2 spike protein appears to work nicely in the tissue culture assay, but lentiviruses incorporate host cell factors in their membrane. Do those play a role in the infectivity of the pseudotyped LV? Also, are the numbers and spatial arrangements of SARS-CoV-2 spike protein on LV comparable to arrangements on real SARS-CoV-2?
Minor point, the authors should explain abbreviations, such as SVF.
Author Response
Dear Editor,
We are pleased to know that our work has been found interesting and useful in current COVID-19 pandemic by the referees. We thank them for all their constructive suggestions and we have made the changes according to their remarks. We also noticed some typos and error in S. Sousa name as well as inversion in the Figure 5D legend which have been now corrected. Please find below our answer to referees comments and suggestions.
REFEREE 1 :
- “The justification for some of the conditions / assumptions used for the simulations is not clear. Ionic strength, charge and solubility near membranes can have a decisive effect on protein-protein interactions, in particular those that result in membrane fusion. Related to that point, a discussion for the parameters that may be important for the in vitro interaction assay will be useful”.
We agree with the referee that this justification was not clearly indicated in the MS. We have added this information in the revised MS (lanes 252-255):
Simulations were run with 0.150 M NaCl in a neutral simulation box subject to periodic boundary conditions, in line with previous molecular dynamics studies on the SARS-COV-2 S1 RDB complexed with ACE2 (the following references have been also added).
- Quantitative determination of mechanical stability in the novel coronavirus spike protein - Rodrigo A. Moreira, Mateusz Chwastyk, Joseph L. Baker, Horacio V. Guzman and Adolfo B. Poma Nanoscale, 2020, 12, 16409-16413 DOI: 10.1039/d0nr03969a
- Molecular Dynamics Reveals Complex Compensatory Effects of Ionic Strength on the Severe Acute Respiratory Syndrome Coronavirus 2 Spike/Human Angiotensin-Converting Enzyme 2 Interaction - Anacleto Silva de Souza, Jose David Rivera, Vitor Medeiros Almeida, Pingju Ge, Robson Francisco de Souza, Chuck Shaker Farah, Henning Ulrich, Sandro Roberto Marana, Roberto Kopke Salinas, and Cristiane Rodrigues Guzzo - The Journal of Physical Chemistry Letters 2020 DOI: 10.1021/acs.jpclett.0c02602
- Elucidating Interactions Between SARS-CoV-2 Trimeric Spike Protein and ACE2 Using Homology Modeling and Molecular Dynamics Simulations - Sakkiah Sugunadevi, Guo Wenjing, Pan Bohu, Ji Zuowei, Yavas Gokhan, Azevedo Marli, Hawes Jessica, Patterson Tucker A., Hong Huixiao, Frontiers in Chemistry 2021 DOI: 10.3389/fchem.2020.622632
These parameters used for developing the simulations were based on data obtained in previous in vitro structural studies where the interaction parameters were found to allow efficient and functional interaction between the two partners. These conditions have been then used for setting up and optimizing the in vitro assays. In order to provide additional information and discussion points about the S/ACE2 interaction in these conditions in these conditions we have now included in the revised MS the detailed characterization of each of the pockets identified, in terms of total SASA, polar SASA, non-polar SASA, volume and main interacting amino acid residues (new Table 1).
- “How comparable are the surfaces used for the in vitro assays with the environment near a membrane where the viral spike protein and ACE2 interact? “
The surfaces used in in vitro assays as BLI or AlphaLISA are based on solid phase which are different from the highly dynamic cellular membrane structures. However, the surfaces used in our assays are typically used for studying protein/protein interaction. Alphascreen, for example, has been broadly used to select drugs acting of protein/protein interfaces and was proved to be an efficient approach for selecting drugs acting also in physiological systems. However, we agree with the referee that these conditions can not fully recapitulate the cellular situation especially in the context of protein embedded in lipid membrane. That is the reason why we wanted in this work to develop complementary systems as in cellulo interaction assays allowing monitoring the S/ACE2 interaction in the membrane context as reported in Figure 7. This last approach, based on cell imaging and flow cytometry, allows the detection and the quantification of the interaction between the soluble S-RBD protein and the ACE2 receptor in its membrane environment providing information about a more physiological association. In complement with the lentiviral infectivity assays, which relies on S/ACE2 interaction in membranous context. While all these approaches would allow providing combined information about this association a system allowing monitoring of the S/ACE2 interaction in a fully membranous viral and cellular context remain to be developed as CryoEM tomography for example which remains a technical challenge and beyond the range of the current work. Before having these approaches ready all the biochemical strategies as AlphaLISA or BLI will help in the better understanding of the molecular determinants of the interactions as well as selecting potential drugs able to disrupt the interfaces between partners. Indeed, BLI is a biophysical system that allow the analyzes of interaction kinetics and, thus, the affinity of the partners which remains difficult in a cellular context. We have mention these points in the revised MS (lane 465).
- The LV pseudotyped with SARS-CoV-2 spike protein appears to work nicely in the tissue culture assay, but lentiviruses incorporate host cell factors in their membrane. Do those play a role in the infectivity of the pseudotyped LV?
Referee is right, lentiviruses incorporate host cell factors in their membrane that may affect the infectivity of the SARS-CoV-2 pseudoyped LV vectors. This is one of the reasons for us to use VSVg pseudotyped LV as control and reference for comparative studies since this virus is assumed to incorporate also cell factors. As shown in Figure 4 infectivity of our SARS-CoV-2 LV relies on the presence of both S and ACE2 indicating that the major determinant for the infectivity of these LVs is the S/ACE2 mediated pathway (ACE2 competition or use of HEK293T cells that do not express ACE2 prevent efficient infectivity). Whether additional cellular factors present in the viral membrane may play a role in the SARS-CoV-2 entry remains to be further investigated as well as the impact of local lipids composition of the membrane. Our work may open the way or help in these studies. We have better discussed this in our revised MS (lanes 562-565).
- “Also, are the numbers and spatial arrangements of SARS-CoV-2 spike protein on LV comparable to arrangements on real SARS-CoV-2?”
This is an interesting and important question that may be addressed using the approaches reported here. Indeed, many structural rearrangements occur during the entry and fusion steps in SARS-CoV-2 S protein as also mentioned by referee 3. Answering this question would need additional heavy complementary experiments beyond the range of this MS as CryOEM tomography to get structural information for the S protein embedded in the LV membrane and in the contact of ACE2 in pre- and post- fusion stages. However, since the SARS-CoV-2 LV entry appeared to rely on the S/ACE2 interaction and is also sensitive to soluble ACE2 competition we could assume that the main structural rearrangements required for viral entry are conserved in our system. This has been mentioned in the revised discussion section (lanes 532-534; 561-568).
- “Minor point, the authors should explain abbreviations, such as SVF.”
We have added several abbreviations that were missing.
Reviewer 2 Report
The paper by Lapalleire et al. describes combined in silico and in vitro approaches for a refined characterization of the entry steps of SARS-CoV-2. This aims to better understand the chemical modulation of the interaction between the viral Spike protein and ACE2 receptor. The paper is mostly methodological and reports accurately the technical procedure applied to unravel the mechanisms, underlying Spike-ACE2 receptor binding. Overall, think this manuscript can be of interest for an improved understanding of SARS-CoV-2 entry.
Some minor points to adress are as follows:
I would clarify better that the experiments were all performed by using the Spike lacking the mutation D614G, currently presents as dominant SARS-CoV-2 specie worldwide, since it is known a different binding property of this mutation respect to ACE-2 receptor binding.
Title: A space should be addedd between moniotring and SARS. Secondly: is it really necessary to use both in vitro and in cellulo? I would suggest to keep only the first.
Line 216 plated on plated should represent a typo, please correct
Line 217 polylysin poly-L-Lysine. It should be sufficient the second poly-L-Lysine
Discussion: Several lines contains words written in a different font format. Please correct this.
Lines 530-532 Please cite the review Artese A et al, DRU, 2020 reporting new potential drugs targeting the interface ACE2-receptor and Spike protein.
“Taken together, all the approaches reported in this work would allow the analysis of the SARS-CoV-2 entry step from the very molecular point to the overall cellular mechanism.” Please rephrase better the sentence.
Author Contributions, Funding, Data Availability Statement, Conflicts of Interest are missing. Please insert this information.
Author Response
Dear Editor,
We are pleased to know that our work has been found interesting and useful in current COVID-19 pandemic by the referees. We thank them for all their constructive suggestions and we have made the changes according to their remarks. We also noticed some typos and error in S. Sousa name as well as inversion in the Figure 5D legend which have been now corrected. Please find below our answer to referees comments and suggestions.
REFEREE 2 :
- “I would clarify better that the experiments were all performed by using the Spike lacking the mutation D614G, currently presents as dominant SARS-CoV-2 specie worldwide, since it is known a different binding property of this mutation respect to ACE-2 receptor binding.”
We agree with the referee that this needs to be mentioned. Indeed all the work has been performed using the “wild type” spike protein. We have mentioned this better along the revised manuscript. All the reported procedures can be now used to study additional variants as D614G, UK, South Africa ones. We have discussed this point in the new discussion (lanes 552-554).
- “Title: A space should be added between monitoring and SARS. Secondly: is it really necessary to use both in vitro and in cellulo? I would suggest to keep only the first.”
We have corrected the typo in the MS title. However, we would like to keep, if possible, the “in cellulo” mention in the title to avoid any misunderstanding for the readers and make them aware of the use of cellular models in our study in addition to biochemical approaches. If the title remains too long we could change it into ‘Molecular and cellular models for monitoring SARS-CoV-2 spike/human ACE2 complex, viral entry and cell fusion”.
- “Line 216 plated on plated should represent a typo, please correct.”; “Line 217 polylysin poly-L-Lysine. It should be sufficient the second poly-L-Lysine”; “Discussion: Several lines contains words written in a different font format. Please correct this.”
All these typos have been corrected.
- “Lines 530-532 Please cite the review Artese A et al, DRU, 2020 reporting new potential drugs targeting the interface ACE2-receptor and Spike protein.”
We agree that this reference was missing and it has been now included in the new MS (lane 566 of the revised MS).
- “Taken together, all the approaches reported in this work would allow the analysis of the SARS-CoV-2 entry step from the very molecular point to the overall cellular mechanism.” Please rephrase better the sentence.
We have modified this sentence.
- “Author Contributions, Funding, Data Availability Statement, Conflicts of Interest are missing. Please insert this information.”
All the information are now part of the revised MS.
Reviewer 3 Report
The study by Lapaillerie et al. shows an interdisciplinary approach quite needed in current COVID-19 pandemic for the study of S-ACE2 interaction responsible which is responsible for the cell entry. The study spands on via a combination of in silico, in vitro and in cellulo models. I find promising the platform and I would appreciate to follow my (major) comments below before accepting for publication. Line 27: I would not call "novel computational model..." as several studies employed the 6M0J file. Please rephrase or remove it. Line 46-48: I would suggest to describe in a more pedagogic manner the process of cell entry. Following standard available literature (https://doi.org/10.1126/science.abd4251, https://doi.org/10.1080/07391102.2020.1758788, https://doi.org/10.1016/j.antiviral.2020.104792). Replication occurs at the later stage. Please add a brief description of it following a scheme: i) first contact with cell mediated by ACE2, ii) conformation of the S protein (close to prefusion state), iii) transition to post fusion state and iv) fusion of the cell-viral membrane and the release of the genetic viral material. Then we have the replication process. Perhaps, here it a good place to add the message of the stability and glycosylation state of the S protein key for cell entry. See my comment at the bottom of this letter. Line 75: As far I see the authors have only employed the RBD in the study (referred by PDB code 6M0J). Please be more specific and replace S1 by RBD. Here and in other instances. Line 78,104,..., and so on: Remove the doi number and replace by the correct citation format Line 79: They author do no mention about the 1 N-glycan in RBD and 2 in ACE, where those taken into account?Line 80: Please replace density by packing Line 83: Leap is a toolkit from amber, please add it. Line 91-97: This is not a standard setup for MD methods. In summary the author write: 1) run1 (NVT, 50 ps), 2) run2 (NVT, 50 ps) and then run3 (NPT,400 ns). In my opinion, there is an step not mentioned (i.e. Energy minimisation-- add the number of steps and method used), then the second step should refers to a short NPT and then production run with 400 ns. Please improve the description. Line 101: Replace RMDs by RMSDLine 102: Replace formed by formationLine 102-107: It is not clear, the authors construct 8 cluster and then select one to carry out the pocket/cavity analysis with Pocket. There is a need to inform which measure (e.g. RMSD with respect to C-alpha atom, a cutoff, global distance test (GDT) , etc) has been used to partition the trajectory and construct the clusters? Line 109: residue segment (438-516) is not in the range of the RBD which is (329-521). Please explain the reason behind the choice. Line 109: remove one bracket in ((HIS Line 235: replace: minimal....(RBD) by receptor binding protein (RBD) of S protein Line 237-239: What is the criterion for the selection of 8 representative structure in 400 ns? Line 243-244: what is the meaning here about "campaign"? I guess the authors refer to test/studies. Line 250-252: The presence of pockets is a very net observation. I suggest to the authors to show more detailed structural information about pockets, e.g. the residues involved in each of the pockets and the determination of the closest HBs to be disrupted, size in nm^2, etc. If not available some tools like: http://www.ifpan.edu.pl/~chwastyk/spaceball/ can be straightforwardly used to find geometric parameters for pockets. Line 255-263: Caption should be improved: For instance, replace complex S-ACE2 by S-ACE2 complex. Also, modelized by modelled. Note that,- Panel A and B seem to present the same information. One can keep one of them, otherwise explain in main text the reasoning for having two (the views are almost identical). There is not description of the colours used (gray for RBD and coloured for ACE2). - Replace in Fig.1C Protein S —> RBD of S, as the full S structure was not used.- What is ProtS-ACE2? Line 268-269: Simiar as above, the region (438-516) does not refer to RBD region which is in the range (329-521). Can the authors justify the use of this region in vitro test, what about inducing unfolding in the RBD domain? Line 318, Line 320: KD, kD, please use same notation Line 468-469: I have not found here or the introduction any reference about the spike S protein stability (for instance include refs. https://doi.org/10.1039/D0NR03969A, and a recent study in MDPI https://doi.org/10.3390/ma13235362) and the glycosilation state (https://doi.org/10.1021/acscentsci.0c01056) which are key for cell entry at the early stage. Line: 479-480: Please replace: ", in addition to" by "which suggest" . Avoid repetition. Line 482: full stop in red colour, change Line 522: replace biophysics by biophysicalAuthor Response
Dear Editor,
We are pleased to know that our work has been found interesting and useful in current COVID-19 pandemic by the referees. We thank them for all their constructive suggestions and we have made the changes according to their remarks. We also noticed some typos and error in S. Sousa name as well as inversion in the Figure 5D legend which have been now corrected. Please find below our answer to referees comments and suggestions.
REFEREE 3 :
- “Line 27: I would not call "novel computational model..." as several studies employed the 6M0J file. Please rephrase or remove it.”
We agree with referee this statement was not correct and was rephrased.
- “Line 46-48: I would suggest to describe in a more pedagogic manner the process of cell entry. Following standard available literature (https://doi.org/10.1126/science.abd4251, https://doi.org/10.1080/07391102.2020.1758788, https://doi.org/10.1016/j.antiviral.2020.104792). Replication occurs at the later stage. Please add a brief description of it following a scheme: i) first contact with cell mediated by ACE2, ii) conformation of the S protein (close to prefusion state), iii) transition to post fusion state and iv) fusion of the cell-viral membrane and the release of the genetic viral material. Then we have the replication process. Perhaps, here it a good place to add the message of the stability and glycosylation state of the S protein key for cell entry. See my comment at the bottom of this letter.”
We agree that the entry step, central point studied in our work, was not described clearly enough. We have better developed this description in the introduction section in order to provide a more pedagogic view of this process as asked by referee (see lanes 51-56). We also introduced the information about the stability of the S protein, its conformational changes and the role of glycans in the discussion section (see lanes 558-562).
- “Line 75: As far I see the authors have only employed the RBD in the study (referred by PDB code 6M0J). Please be more specific and replace S1 by RBD. Here and in other instances.”
Referee is correct since all the in vitro analyzes were performed using the RBD domain of S1. We have better mentioned this along the MS and modified the text when required.
- “Line 78,104,..., and so on: Remove the doi number and replace by the correct citation format”
Sorry for these typos, we have replaced the DOI numbers by their corresponding citations.
- “Line 79: They author do no mention about the 1 N-glycan in RBD and 2 in ACE, where those taken into account?”
We understand the reviewer’s point. A total of 5 N-glycan sites were considered, as present in the 6M0J structure (N53, N90, N322 and N546 in ACE2 and N343 in S1 RDB). This information has now been clarified in the methods section (lanes 86-88).
- “Line 80: Please replace density by packing “.
Following the reviewer suggestion we have replaced density by packing.
- “Line 83: Leap is a toolkit from amber, please add it.”
Following the reviewer suggestion we included this information regarding Leap.
- “Line 91-97: This is not a standard setup for MD methods. In summary the author write: 1) run1 (NVT, 50 ps), 2) run2 (NVT, 50 ps) and then run3 (NPT,400 ns). In my opinion, there is an step not mentioned (i.e. Energy minimisation-- add the number of steps and method used), then the second step should refers to a short NPT and then production run with 400 ns. Please improve the description.” .
Following the reviewer recommendation we have improved the description of the Molecular dynamics stage. The protocol employed involved three main states: (1) Minimization; (2) Equilibration, including heating from 0 to 310 K (NVT); (3) Production (NPT). All stages were performed with the AMBER18 software and with the AMBER14SB force field.
Energy Minimization was performed along 4 sequential stages. On stage 1, only the water molecules were optimized (total 2500 steps, 1250 steps with the steepest descent algorithm - SDA and 1250 with the conjugated gradient algorithm - CGA), with the rest of the system remaining fixed with a harmonic potential of 50 kcal mol-1 Å-2; On stage 2, all the hydrogen atoms of the system were optimized (2500 steps, 1250 SDA + 1250 CGA), with the rest of the system remaining fixed with a harmonic potential of 50 kcal mol-1 Å-2. On stage 3, only the backbone atoms (CA and N) were kept restrained with a harmonic potential of 50 kcal mol-1 Å-2, while the positions of the remaining atoms of the system (particularly the amino acid side chains) were energy minimized (2500 steps, 1250 SDA + 1250 CGA). Finally, on stage 4, all the system was energy minimized without restrains for 10,000 steps (5000 SDA + 5000 CGA).
Equilibration. Equilibration was divided into 2 stages, each performed along 50 ps in an NVT ensemble. On stage 1 of the equilibration the temperature was gradually increased from 0 to 310 K along the full 50 ps of the simulation. On stage 2 of the equilibration, the system was further equilibrated, already at the final temperature, along additional 50 ps of the simulation.
This division of the equilibration run into two stages, has become quite common in recent years as it ensures a more gradual heating procedure and in general a more careful equilibration, especially for difficult cases. In older set-ups (5-10 years ago), we and other authors, typically performed one single equilibration NVT stage with a typical length of 50-100 ps, with a faster heating. These older set-ups have been progressively substituted.
Hence, the present protocol is conceptually superior. However, for a typical biomolecular system such of the one targeted in the present study, both alternatives would result in neglectable differences.
Production. The production run was performed along 400 ns in an NPT ensemble. The initial nanoseconds of the MD simulations are normally regarded as an adjustment of the system to the NPT ensemble and are not considered for the determination of dynamic properties from the trajectory.
We thank the reviewer for this opportunity to clarify the MD protocol.
- “Line 101: Replace RMDs by RMSD”; “Line 102: Replace formed by formation.”
This has been corrected
- “Line 102-107: It is not clear, the authors construct 8 cluster and then select one to carry out the pocket/cavity analysis with Pocket. There is a need to inform which measure (e.g. RMSD with respect to C-alpha atom, a cutoff, global distance test (GDT) , etc) has been used to partition the trajectory and construct the clusters?”.
Some details were missing regarding the application of the clustering method. These have now been included:
- “The cluster analysis tool in cpptraj was used to partition the trajectory into 8 clusters, based on RMSD relative to the backbone C-alpha carbons, using the average-linkage hierarchical agglomerative approach with a 2.0 Å cut-off.”
In addition, following the cluster analysis, 1 representative structure per cluster was determined and used for the cavity detection studies (total 8 clusters -> 8 structures; plus the X-ray structure). This aspect has now been also clarified in the revised manuscript.
- “Line 109: residue segment (438-516) is not in the range of the RBD which is (329-521). Please explain the reason behind the choice”.
We agree with the referee we did not explain clearly this choice in the MS. Based on previous studies as Lan et al., Nature 2020 and Wrapp et al Science 2020 the S 438-516 fragment includes the minimal receptor binding motif (RBM, 438-506) that interacts directly with ACE2. In order to focus the work on this minimal interface we used the 438-516 fragment. We have mentioned this better in the revised MS.
- “Line 109: remove one bracket in ((HIS Line 235: replace: minimal....(RBD) by receptor binding protein (RBD) of S protein”.
We have now corrected these points.
- “Line 237-239: What is the criterion for the selection of 8 representative structure in 400 ns?”
The criterium for the choice of 8 representative structures (one per each cluster) was RMSD of the alfa carbons. From each of the 8 clusters identified in the cluster analysis, the structure closest to the corresponding centroid was automatically assigned by cpptraj during the cluster analysis. We have now clarified this information in the methods section.
- “Line 243-244: what is the meaning here about "campaign"? I guess the authors refer to test/studies.”
We have now replaced “campaign” by “study” as to make the sentence clearer, in line with the reviewer suggestion.
- “ Line 250-252: The presence of pockets is a very net observation. I suggest to the authors to show more detailed structural information about pockets, e.g. the residues involved in each of the pockets and the determination of the closest HBs to be disrupted, size in nm^2, etc. If not available some tools like: http://www.ifpan.edu.pl/~chwastyk/spaceball/ can be straightforwardly used to find geometric parameters for pockets.”
We agree with the reviewer. We have now included the detailed characterization of each of the pockets identified, in terms of total SASA, polar SASA, non-polar SASA, volume and main interacting amino acid residues. Values were obtained with FPocket.
Table 1. Characterization of the binding pockets identified on the X-ray structure and on the 8 representative structures obtained from the cluster analysis in terms of SASA (solvent accessible surface area), volume and main interacting amino acid residues at the ACE2 and S1 RBD interface. Values obtained with FPocket.
|
Cluster |
|
Total SASA (Å2) |
Polar SASA (Å2) |
Non-Polar SASA (Å2) |
Volume (Å3) |
ACE2 Main Residues |
S1 RDB Main Residues |
|
X-Ray |
1 |
141.0 |
75.8 |
65.2 |
351.1 |
Asn33, His34, Glu37, Lys353 |
Arg403, Lys417, Tyr453, Tyr495, Tyr505, |
|
|
2 |
193.8 |
98.4 |
95.4 |
738.9 |
Gln42, Leu45, Asn49, Asn61, Asn64, Lys68 |
Gly446, Tyr449 |
|
|
3 |
241.4 |
115.8 |
125.6 |
827.5 |
Thr324, Phe356, Arg393 |
Gly502, Val503, Tyr505 |
|
Cluster 1 |
1 |
223.5 |
134.1 |
89.4 |
828.7 |
Asn33, His34, Glu37, Asp38, Lys353, Pro389, Phe390 |
Arg403, Glu406, Lys417, Ile418, Tyr495, Tyr505 |
|
|
2 |
155.7 |
79.6 |
76.1 |
543.4 |
Phe28, Lys31, Phe32, Glu35, Asp38, Leu39, Phe72, Gln76, Leu79 |
Glu484, Phe486, Asn487, Tyr489 |
|
|
3 |
152.0 |
43.3 |
108.7 |
525.8 |
Thr324, Gln325, Asn330, Asp355, Phe356 |
Val503, Tyr505 |
|
Cluster 2 |
1 |
304.7 |
173.1 |
131.6 |
919.7 |
His34, Glu37, Asp38, Lys353, Pro389, Arg393 |
Arg403, Gln409, Lys417, Ile418, Tyr495 |
|
|
2 |
174.4 |
112.2 |
62.2 |
549.8 |
Lys31, Gln76, Leu79 |
Glu484, Tyr489, Phe490 |
|
Cluster 3 |
1 |
207.3 |
98.6 |
108.7 |
603.1 |
Lys353, Phe356, Arg393 |
Asp405, Tyr505 |
|
|
2 |
257.4 |
142.6 |
114.7 |
750.6 |
Thr324, Gln325, Glu329, Asp355 |
Thr500, Gly502, Val503
|
|
Cluster 4 |
1 |
213.1 |
128.6 |
84.5 |
642.8 |
Asn33, His34, Pro389, Arg393 |
Arg403, Asp405, Lys417 |
|
Cluster 5 |
1 |
232.1 |
122.2 |
109.9 |
784.6 |
Asn33, His34, Pro389, Phe390 |
Asp405, Lys417, Ile418, Tyr495, Tyr505 |
|
Cluster 6 |
1 |
155.8 |
96.6 |
59.2 |
535.4 |
Asp30, Asn33, His34, Glu37, Asp38, Lys353, Phe390 |
Arg403, Lys417, Tyr495, Tyr505 |
|
|
2 |
187.7 |
75.4 |
112.3 |
566.0 |
Asp350, Phe356, Tyr385, Arg393 |
Gly502, Tyr505 |
|
Cluster 7 |
1 |
175.3 |
107.7 |
67.6 |
565.9 |
Glu37, Asp38, Leu39, Lys353 |
Leu452, Arg403, Gln493, Tyr495, Tyr505 |
|
|
1 |
285.5 |
128.5 |
157.0 |
833.7 |
Asn33, Thr324, Gln325, Phe356, Pro389, Arg393 |
Arg403, Asp405, Val503 |
|
Cluster 8 |
1 |
252.0 |
144.5 |
107.5 |
799.2 |
Thr20, Glu23, Gln24, Lys26, Thr27, Asp30 |
Lys417, Tyr421, Phe456, Tyr473, Ala475 |
|
|
|
|
|
|
|
|
|
|
|
|
|
|
|
|
|
|
- “ Line 255-263: Caption should be improved: For instance, replace complex S-ACE2 by S-ACE2 complex. Also, modelized by modelled. Note that,- Panel A and B seem to present the same information. One can keep one of them, otherwise explain in main text the reasoning for having two (the views are almost identical). There is not description of the colours used (gray for RBD and coloured for ACE2). – Replace in Fig.1C Protein S —> RBD of S, as the full S structure was not used.- What is ProtS-ACE2?”
We have done all the changes suggested by the reviewer, in terms of the caption and of the figure itself
- “Line 268-269: Simiar as above, the region (438-516) does not refer to RBD region which is in the range (329-521). “
This has been now explained. This fragment has been reported to interact directly with ACE2 and, thus, is functionally structured (Lan et al., Nature 2020 and Wrapp et al Science 2020)
- ‘Line 318, Line 320: KD, kD, please use same notation’
We have homogenized our notation.
- “Line 468-469: I have not found here or the introduction any reference about the spike S protein stability (for instance include refs. https://doi.org/10.1039/D0NR03969A, and a recent study in MDPI https://doi.org/10.3390/ma13235362) and the glycosilation state (https://doi.org/10.1021/acscentsci.0c01056) which are key for cell entry at the early stage.”
Referee is right these points were missing. We have now discussed them in the revised MS (lanes 558-562).
- “Line: 479-480: Please replace: ", in addition to" by "which suggest" . Avoid repetition. Line 482: full stop in red colour, change L, line 522: replace biophysics by biophysical”
All these points have been corrected.
Round 2
Reviewer 3 Report
The authors have improved the manuscript according with high academic standards and I sincerely support the publication of this MS. I wish them luck in future endeavours.
Small changes:
line 94: change RDB by RBD
line 313: change 0.150M to 0.15M
line 324: Change SARS-COV-2 RDB to SARS-CoV-2 RBD
line 657: "Indeed, adding increasing concentrations of soluble..." --> I think there is not need of "adding".